# Neutron Total Scattering Investigation of the Dissolution Mechanism of Trehalose in Alkali/Urea Aqueous Solution

**DOI:** 10.3390/molecules27113395

**Published:** 2022-05-25

**Authors:** Changli Ma, Taisen Zuo, Zehua Han, Yuqing Li, Sabrina Gärtner, Huaican Chen, Wen Yin, Charles C. Han, He Cheng

**Affiliations:** 1Institute of High Energy Physics, Chinese Academy of Sciences (CAS), Beijing 100049, China; machangli@ihep.ac.cn (C.M.); zuots@ihep.ac.cn (T.Z.); hanzh@ihep.ac.cn (Z.H.); liyuqing@ihep.ac.cn (Y.L.); chenhuaican@ihep.ac.cn (H.C.); yinwen@ihep.ac.cn (W.Y.); 2Spallation Neutron Source Science Center, Dongguan 523803, China; 3University of Chinese Academy of Sciences, Beijing 100049, China; 4STFC ISIS Facility, Rutherford Appleton Laboratory, Didcot OX11 0QX, UK; sabrina.gaertner@stfc.ac.uk; 5Institute for Advanced Study, Shenzhen University, Shenzhen 508060, China; han.polymer@gmail.com

**Keywords:** neutron total scattering, cellulose, dissolution mechanism, layered structure, complexation

## Abstract

The atomic picture of cellulose dissolution in alkali/urea aqueous solution is still not clear. To reveal it, we use trehalose as the model molecule and total scattering as the main tool. Three kinds of alkali solution, i.e., LiOH, NaOH and KOH are compared. The most probable all-atom structures of the solution are thus obtained. The hydration shell of trehalose has a layered structure. The smaller alkali ions can penetrate into the glucose rings around oxygen atoms to form the first hydration layer. The larger urea molecules interact with hydroxide groups to form complexations. Then, the electronegative complexation can form the second hydration layer around alkali ions via electrostatic interaction. Therefore, the solubility of alkali aqueous solution for cellulose decreases with the alkali cation radius, i.e., LiOH > NaOH > KOH. Our findings are helpful for designing better green solvents for cellulose.

## 1. Introduction

Cellulose is the world’s most produced natural polymer, and it is a potential candidate to replace petroleum-based materials. To achieve high performance, cellulose has to be dissolved first [1].

Cellulose is difficult to dissolve. Strong inter- and intra-chain interactions prevent its structure from being deconstructed. In industry, the widely used viscose method produces a large amount of alkaline and acidic waste, carbon disulfide and hydrogen sulfide gases. It pollutes the environment. N-methylmorpholine noxide (NMMO) and ionic liquids (ILS) are environment friendly, but they have high costs [1].

Around 2000, Prof. Zhang proposed the use of precooled NaOH/LiOH urea aqueous solution to dissolve cellulose [1,2]. This solvent has the advantages of fast dissolution, low cost and low pollution and, thus, has good application prospects. Unfortunately, the solubility of cellulose in green solvent is still too low to reach the requirements of industrial production. We need to know the dissolution mechanism first to increase its solubility. Lots of methods, such as NMR, FTIR, DSC, TEM, et al., have been used. These studies qualitatively showed that cationic hydrates are more easily adsorbed around cellulose molecules to form a new, stable hydrogen bond network at low temperatures, and hydrates of urea molecules form a sheath-like inclusion complex (IC) around their periphery [3,4,5]. Bjorn et al. summarized the dissolution mechanism of cellulose and proposed that cellulose is amphiphilic and that hydrophobic interactions are important for its solubility [6,7,8,9]. Wolfgang et al. agreed that hydrophobic interactions are the driving forces in an amorphous system, but all types of cellulose are highly crystalline so hydrogen bonds must be broken before hydrophobic interactions can be effective, and this requires a strong alkaline medium [10]. To reveal the dissolution mechanism of cellulose in alkaline/urea aqueous solution, a detailed atomic picture needs to be presented.

The combination of neutron total scattering and empirical potential structure refinement (EPSR) can observe in situ the most probable all-atom structure of the liquids [11,12]. In a previous study, we used the combined methods to study the atomic structure of trehalose in NaOH/urea aqueous solution [13]. We used trehalose as the model molecule for cellulose, because it has similar glucose rings and is one of the disaccharide molecules that has no reducibility (other binary sugars, such as cellobiose, glucose, maltose and lactose, are oxidized in the alkaline solution during neutron total scattering experiments). We found that NaOH, urea and water work cooperatively to dissolve trehalose. Na^+^ accumulates around electronegative oxygen atoms in the hydration shell, while urea molecules only participate in the dissolution process via Na^+^ bridging. Additionally, we predicted that alkali with smaller ions, such as LiOH, have better solubility for cellulose.

To prove this, we further observed the microscopic dissolution pictures in two different alkali/urea aqueous solutions, i.e., LiOH and KOH. Then, we compared them with previous results in NaOH/urea aqueous solution. The hydration shell of trehalose has a layered structure; cations directly interact with the glucose rings to form the first hydration layer. It destroys their intra- and intermolecular hydrogen bonds. Thus, the smaller the radius of the cation, the easier it approaches the inside of the glucose ring. From K^+^ to Na^+^ to Li^+^, its ability to dissolve cellulose gradually increases. Urea does not directly interact with glucose rings, and it forms strong complexations with hydroxide groups. The urea hydration complexation forms the second hydration layer via electric interaction. It prevents it from re-aggregating. Temperature effect was also investigated. The atomic structure of the solution did not change when it was cooled down to −10 °C (or −5 °C). Taking into account the fact that cellulose only dissolves in green solvent at lower temperatures, the dissolution had to be a dynamic process.

## 2. Theory and Methods

### 2.1. Neutron Scattering Method and SANDALS

The total neutron scattering experiment is an important method for the study of the atomic structure of liquids. In neutron scattering, the observed neutron structure factor (*F(Q)*) and the atomic structure of the sample have the relationship:(1)$$F\left(Q\right)=\overset{N}{\underset{\alpha =1}{\sum }}c_{\alpha }b_{\alpha }^{2}+\underset{\alpha =1,\beta \geq \alpha }{\sum }\left(2-\delta_{\alpha \beta }\right)c_{\alpha }c_{\beta }b_{\alpha }b_{\beta }\left\{4\pi \rho \int_{0}^{\infty }r^{2}\left(g_{\alpha \beta }\left(r\right)-1\right)\frac{\sin\left(Qr\right)}{Qr}dr\right\}\;$$
where *Q* and *r* are the momentum transfer and the distance between the two atoms in the sample; *c_i_* and *b_i_* (*i =*
*α*, *β*) are the quantity ratio and neutron scattering length of *i* species; *δ_αβ_* is the Kronecker *δ* function; *ρ* is the atomic number density of the sample; and *g**_αβ_ (r)* is the radial distribution function (hereinafter referred to RDF), which reflects the microscopic atomic structure of amorphous matter.
(2)$$g_{\alpha \beta }\left(r\right)=\frac{n_{\alpha \beta }}{4\pi r^{2}\;dr\rho_{\beta }}\;$$
where *n**_αβ_* is an average number of β atoms around an *α* atom contained in a spherical shell with *r* radius and *dr* thickness; *ρ_β_* is the average number density of *β* atoms in the sample; and *g**_αβ_(r)* describes the number density change of the *β* atom as the function of the distance from the *α* atom, which reflects the interaction between atoms *α* and *β*. We used *g_(1)_**_αβ_(r)* to mark the first peak of *g**_αβ_(r)*. It represents the closest, most probable distance between atoms *β* and *α*. The peak height indicates the ratio of the atomic number density of atom *β* to its average value at this position.

The neutron total scattering experiments were performed at SANDALS in ISIS (Didcot, UK) and Multi-Physics Instrument (MPI) in CSNS (Dongguan, China). The scattering vector (Q) range was from 0.1 Å^−1^ to 50 Å^−1^. This means that the instrument could only measure the micro-structure of samples from ~0.1 Å to ~30 Å. Therefore, we chose trehalose as the model molecule for cellulose. The size of trehalose is ~11.6 Å; we could, thus, use a neutron total scattering instrument to observe its microscopic atomic structure in alkali/urea aqueous solution.

### 2.2. EPSR Simulations

The empirical potential structure refinement (EPSR) is a program developed by Prof. Soper to explore the most probable, all-atom structure of an experimental sample based on neutron scattering [11]. EPSR is essentially a Monte Carlo simulation program. In EPSR, there are two kinds of potential energy: reference potential and experimental potential. The reference potential is taken from the molecular dynamics simulation force field, which is used to realize the basic structural constraints of the simulation system, such as molecular structure, the minimum distance between atoms, etc.; the experimental potential is the structural data observed in the neutron scattering experiment [11,12]. After the experiment, we used EPSR simulation to reconstruct the atomic structure of the experimental sample.

### 2.3. Experiment Samples

The trehalose, LiOH, KOH and urea used in the experiment were purchased from Shanghai Aladdin Reagent Company. The purity of trehalose was over 99%, and the purity of alkali and urea was over 99.9% for both. Both deuterated urea and heavy water were purchased from Sigma-Aldrich China. The deuteration rate of heavy water was over 99.9%, and the deuteration rate of urea was over 98%.

In this study, we used SANDALS and MPI to observe 4 kinds of experimental sample with different chemical components. They were aqueous solutions of trehalose/LiOH/urea, trehalose/KOH/urea, trehalose/LiOH and trehalose/KOH. The molar ratio of each component in the sample was consistent with that in Prof. Zhang’s experiment [2]. Each kind of solution included three different deuterated ratio test samples, i.e., full hydrogen, full deuterium and half deuterium. All samples were measured at room temperature (25 °C) and −10 °C. However, the trehalose LiOH/urea aqueous solution froze at −10 °C in the neutron scattering experiment, so we raised the temperature of this sample to −5 °C. Each sample was measured in a flat sample cell of titanium–zirconium alloy with a capacity of 1.3 mL for approximately 6 h. Empty sample cells were measured for approximately 4 h for background subtraction. The sample composition and symbols are shown in Table 1.

For the EPSR simulation and the following discussion of the results, the symbols of the atoms in trehalose were as shown in Figure 1. Oxygen and hydrogen atoms of H_2_O were labeled as OW and HW, respectively; alkali cations were labeled as Li, Na and K, respectively; the oxygen and hydrogen atom of hydroxide anion were labeled as OOH and HOH, respectively; and the carbon, oxygen, nitrogen and hydrogen atoms in the urea molecule were labeled as CU, OU, NU and HU, respectively. The label of each atom and the parameters of reference potential used in EPSR simulation are listed in Table 2(1,2). Among them, epsilon and sigma are the parameters of the Lennard-Jones reference potential (L-J), and q is the charge of the atoms [13,14,15,16,17]. The information for the EPSR simulation box is shown in Table 3. The data about NaOH aqueous solutions can be found in our previous study [13].

## 3. Results

The neutron scattering structure factor and EPSR simulation results of different samples are shown in Figure 2. As shown in the figure, the EPSR simulation results and the neutron scattering profiles were in good agreement. Therefore, the structure reconstructed by EPSR represented the most probable atomic structure of the solutions.

EPSR gave us the most probable, all-atom structures in trehalose alkali/urea aqueous solution. The simulation, as shown in Figure 3, qualitatively showed the distribution diagram of Li^+^, Na^+^, K^+^ ions and urea molecules around trehalose. In order to improve the statistics, the ions and atoms in the figure were the number of atoms gathered in the 100-frame EPSR simulation conformation. Three things could be seen directly. The first thing was that those ions and urea molecules were anisotropically distributed around trehalose because of steric repulsion, so we built a coordinate system around its O1 atom (Figure 1). Thus, RDF could be calculated at different space angles. The second thing was that ions concentrated around the hydrophilic oxygen atoms of trehalose, and smaller ions were much more easily penetrated into the glucose ring, from K^+^ to Na^+^ to Li^+^ (Figure 3a). The third thing was that larger urea molecules could not directly interact with trehalose, and they were further away from trehalose than the ions (Figure 3b).

Then, we quantitatively analyzed the distribution of alkali, urea and water, as shown in Figure 3. We paid specific attention to the hydration shell around the oxygen atoms (we call them Os hereafter) of the glucose rings. They included all of the hydrophilic atoms on the glucose rings.

Because Os are electronegative, the first hydration shell had to be electropositive. There are three kinds of electropositive atom in an alkaline system, i.e., hydrogen atoms of urea and water and cations of alkali. Thus, there were 3 × 4 = 12 kinds of g(r) between them and the Os. There was only one g_(1)_(r) larger than 1, i.e., it was between cation and the Os. From the previous study, we knew that cations play a role in breaking the intra- and inter-molecular hydrogen bonds of cellulose. So, we first examined the RDF distribution between the cations and oxygen atoms of trehalose. Figure 4 compares the RDFs of Li^+^/Na^+^/K^+^ ions and O1~O4. Here, we separated Os into two groups. O1 and O2 were in one group, while O3 and O4 were in the other group. We discussed their RDFs with cations, respectively. In group one, both O1 and O2 were inside the glucose rings. g_(1)O1-K_(r)~5, centered at r1~2.6 Å. This means that the first K^+^ shell around O1 was centered about 2.6 Å, where the K^+^ concentration was five times larger than the average K^+^ concentration in the solution. g_(1)O1-Na_(r)~10, located at r1~2.4 Å, and g_(1)O1-Li_(r)~20 was at r1~2.0 Å. Thus, smaller cations were more close to O1, and their concentrations were 10 and 20 times their averaged concentration in solution. Similar things happened around O2. In group two, both O3 and O4 were in the periphery of the glucose rings. The spaces around them were, thereby, relatively open. g_(1)O-K_(r)~1.8, centered at r1~2.8 Å; g_(1)O-Na_(r)~1.9, centered at r1~2.45 Å; and g_(1)O3-Li_(r)~1.4, centered at r1~2.1 Å. Larger ions more easily concentrated around them.

From these comparisons, it was found that the Li^+^ ions were closest to the two oxygen atoms inside the glucose ring, i.e., O1 and O2, while larger ions, such as Na^+^ and K^+^, were more likely to be enriched near O4. Because the LiOH urea aqueous solution had the most powerful solubility, these comparisons confirmed that the cations in the alkaline solution directly interact with the glucose ring. An interesting result was that K^+^ ions with a relatively larger ionic radius were more concentrated around oxygen atoms away from the center of the glucose ring. This may explain why the KOH/urea aqueous solution had a better dissolving effect on chitin [18]. We suppose that, in the dissolution process of chitin, breaking the association between the side groups may be crucial, while, in the dissolution process of cellulose, breaking the interaction between the glucose backbones is a prerequisite.

Then, we looked for the components in the second layer of the hydration shell. The first hydration shell was electropositive, so the second shell had to be electronegative. There were four electronegative solvent atoms in the system, i.e., OW, NU, OU and OOH. They themselves or their complexations composed the second hydration layer. We listed all of their g(r) with the Os and compared their structures with g_O1-Li_(r) in LiOH/urea aqueous solution. Figure 5 assumes that they were isotropically distributed. We added two dash lines to indicate their possible boundary. It is easy to see that all of those g(r), except g_(1)OW-O3 /O4_(r), were smaller than 1. Therefore, three conclusions can be made, i.e., the second hydration layer almost did not exist; it was anisotropically distributed, or it only existed around O3 and O4 on the periphery of the glucose ring. We checked them one by one.

As shown in Figure 3, the distributions of cations and urea around O1 and O2 were very asymmetric. To quantitatively study the asymmetry of cationic and electronegative atoms around O1, we calculated their g(r) as a function of spatial angle. In the calculation, the setting of the coordinate axis was as shown in Figure 1. We first set a cone with an apex angle of 60° and the Y coordinate axis as the axis of symmetry. Then, we placed the apex of this cone at O1. Finally, we let the cone take O1 as the center of rotation and rotated it around the *X*-axis and calculated the g(r) changes with the rotation angle. The calculation results showed that g_(1)O1-Li_(r) reached its maximum value between 90° and 180°, which coincided with the *Z*-axis. Figure 6 shows g_O1-Li_(r), g_O1-OW_(r), g_O1-OU_(r), g_O1-NU_(r) and g_O1-OOH_(r) along the *Z*-axis. g_(1)O1-OW_(r), g_(1)O1-OOH_(r) and g_(1)O1-NU_(r) were larger than 1.0. Therefore, electronegative OW, OOH, NU or their complexation composed the second hydration layer.

The existence of urea is crucial for dissolving cellulose in green solvent. Previous NMR observations claimed that the amino group of urea forms an electronegative complexation with OH^−^ through hydrogen bonding. Such complexations wrap around cations and eventually form cellulose–NaOH–urea–H_2_O inclusion complexes (ICs) [5,19]. To clarify this, we listed all of the possible complexations between OH^−^ and urea (Figure 7a). There were four possibilities, i.e., HU as the proton donor to form a hydrogen bond with OW (HB1) and OOH (HB2), NU as the proton acceptor (HB3) and OU as the proton acceptor (HB4). Figure 7b gives the RDFs between OW and NU, OW and HU and HW and NU. We used 2.9 ± 0.3Å as the donor–acceptor distance constraint and linear bond (±20°) as the angle constraint to define the hydrogen bond. The possibility of forming a hydrogen bond for HB1 and HB3 was 78.5% and 10.9%, respectively. Thus, HU prefers forming a hydrogen bond with OW. Figure 7c shows the RDFs between OOH and NU, OOH and HU and HOH and NU. If we used 2.5 ± 0.4 Å as the donor–acceptor distance constraint and (±20°) as the angle constrain to define the hydrogen bond, the possibility of forming a hydrogen bond for HB2 was 51.9%. Thus, HU can also form a hydrogen bond with OOH. Finally, Figure 7d is the RDF between OW and OU and HW and OU. They could form a hydrogen bond (HB4), but they were away from the second hydration layer, as shown in Figure 6. Therefore, as a proton donor, the amino group of urea can hydrogen bond with hydroxy group to form complexations. The electronegative complexation forms the second hydration layer.

Temperature effect on dissolution was also investigated. Figure 8 shows the neutron scattering profiles of trehalose in LiOH urea aqueous solutions at 25 °C and −5 °C and their differences. It can be seen that there was no obvious difference between the two different temperatures. Therefore, low temperature does not change the atomic structure of the solutions. It only increases the stability of the solution, making it difficult for the dissolved cellulose to re-aggregate.

## 4. Conclusions

In this research work, we used trehalose as a model molecule and neutron scattering and EPSR simulation as the main tools to study the rapid dissolution mechanism of cellulose in alkali/urea aqueous solution. The three-dimensional atomic structures of trehalose in three different alkali/urea aqueous solutions were thus compared. Alkali, urea and water work cooperatively to dissolve trehalose. A layered hydration shell is crucial. Cations directly interact with the Os of the glucose rings. They first break the inter- and intra-hydrogen bonding. The smaller the ions, the more easily they penetrate into the glucose rings. Urea molecules are too large to approach the glucose ring. As proton donor, their amino group can form a hydrogen bond with the hydroxyl group. The resultant electronegative complexation constitutes the second hydration layer via bridging force. They further stabilize trehalose and prevent it from re-aggregating.

We also found an interesting phenomenon, that is, Li^+^ ions are more concentrated around O1 and O2, while K^+^ ions are more concentrated in the vicinity of O3 and O4, especially near O4. Although the low-temperature aqueous solution of KOH urea could not dissolve cellulose, it could dissolve chitin; at the same time, the low-temperature aqueous solution of LiOH/NaOH urea could dissolve cellulose, but could not dissolve chitin [18]. Based on this experimental evidence, we speculate: in the dissolution of cellulose, breaking the hydrogen bond formed by O1 and O2 is the key factor, while, in the dissolution of chitin, breaking the hydrogen bonds formed by its hydroxyl and amide groups is crucial. These findings are expected to be verified in future experiments.

## Figures and Tables

**Figure 1 molecules-27-03395-f001:**
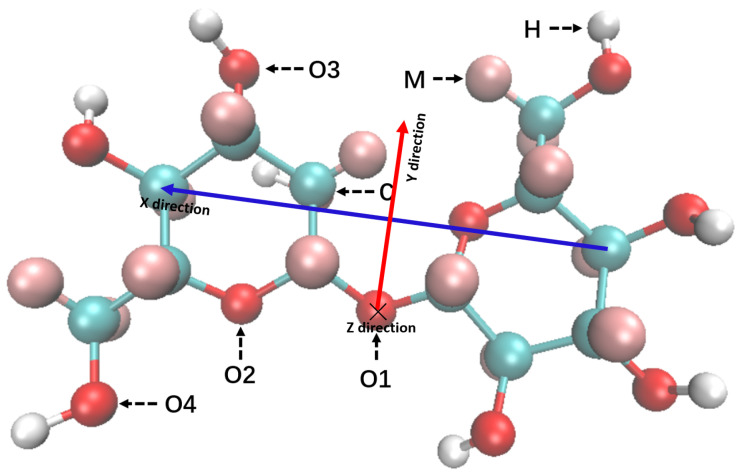
Structure of a trehalose molecule and its atomic labels used in EPSR simulation. All carbon atoms are labeled as C; all hydrogen atoms connected to the oxygen atom are labeled as H; the remaining hydrogen atoms connected to the carbon atom are labeled as M; the oxygen atom linking the two glucose rings is labeled as O1; the oxygen atoms on the glucose ring is labeled as O2; the oxygen atoms on the hydroxyl group connected to the glucose ring are labeled as O3; and the oxygen atom on the methyl group is labeled as O4. In order to describe the distribution of atoms around trehalose with spatial angles, we set up a coordinate system with O1 as the coordinate origin. The blue arrow connecting two C atoms at the glucose rings represents the direction of the *X*-axis; the red arrow pointing vertically to the blue arrow from O1 is the direction of the *Y*-axis; the *Z*-axis is the cross product of the *X*-axis and the *Y*-axis, which is approximately perpendicular to the paper.

**Figure 2 molecules-27-03395-f002:**
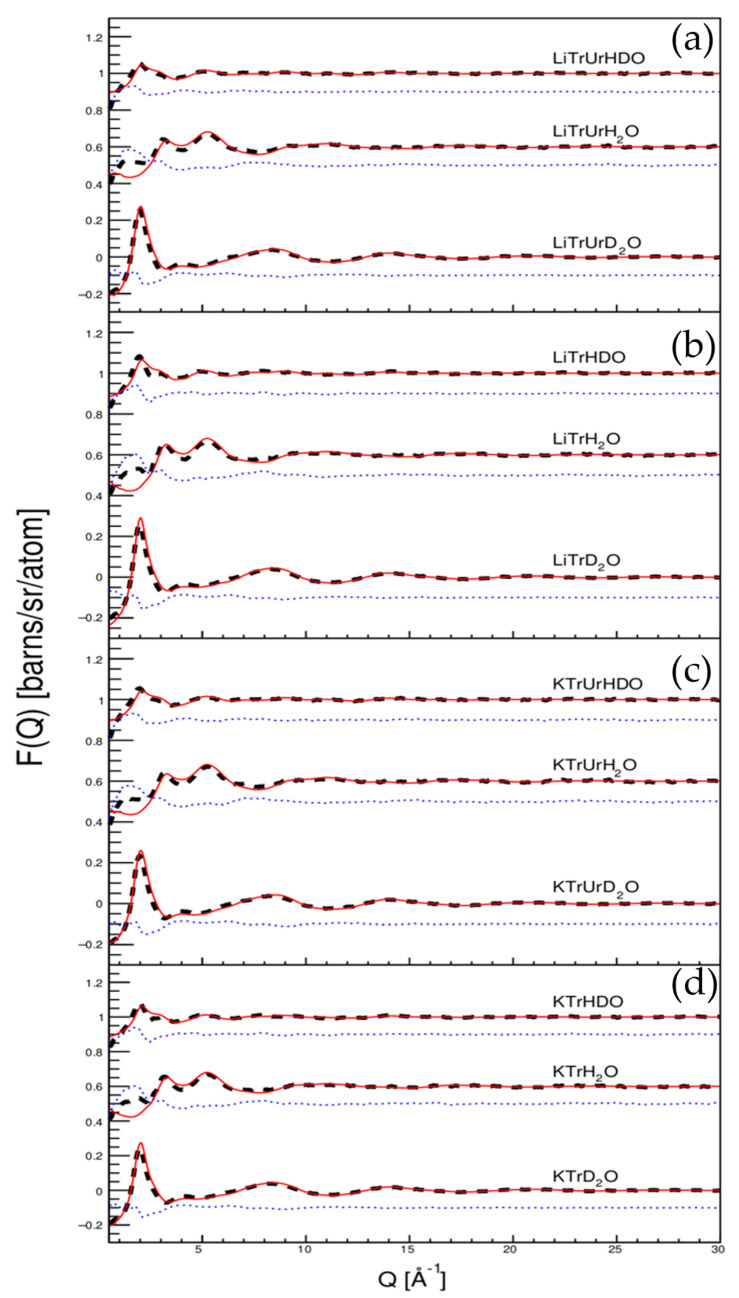
Experimental neutron total scattering profiles (dashed black lines), the EPSR fitted neutron structure factors (red lines) and their differences (blue dots). (**a**) Trehalose in LiOH/urea aqueous solution, (**b**) trehalose in LiOH aqueous solution, (**c**) trehalose in KOH/urea aqueous solution, (**d**) trehalose in KOH aqueous solution. Data have been offset for clarity.

**Figure 3 molecules-27-03395-f003:**
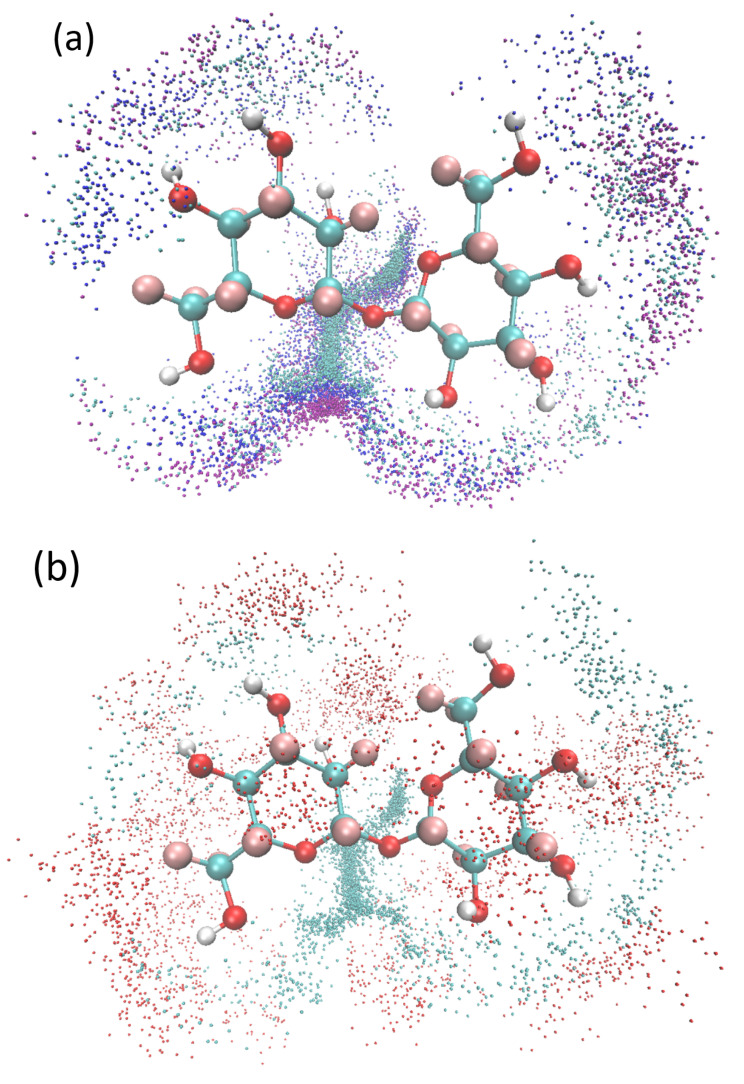
The schematic diagram of the accumulation of alkali metal ions and urea molecules around trehalose. (**a**) Li^+^ (blue dots), Na^+^ (dark blue dots), K^+^ (red dots) ions are distributed around a trehalose molecule. (**b**) Distribution of Li^+^ and the oxygen atoms of urea (red dots) around a trehalose molecule.

**Figure 4 molecules-27-03395-f004:**
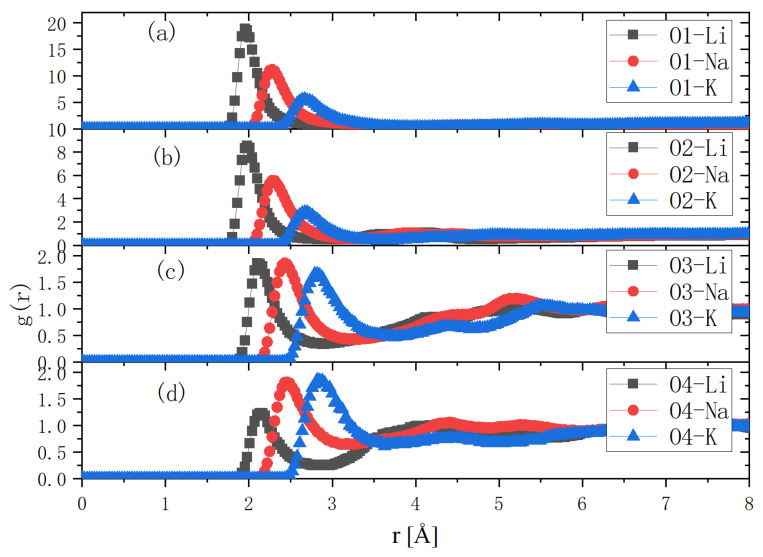
RDF distributions between the alkali metal ions and the solute oxygen atoms. g(r)s between Li^+^, Na^+^, K^+^ and O1 (**a**), O2 (**b**), O3 (**c**), O4 (**d**) of trehalose.

**Figure 5 molecules-27-03395-f005:**
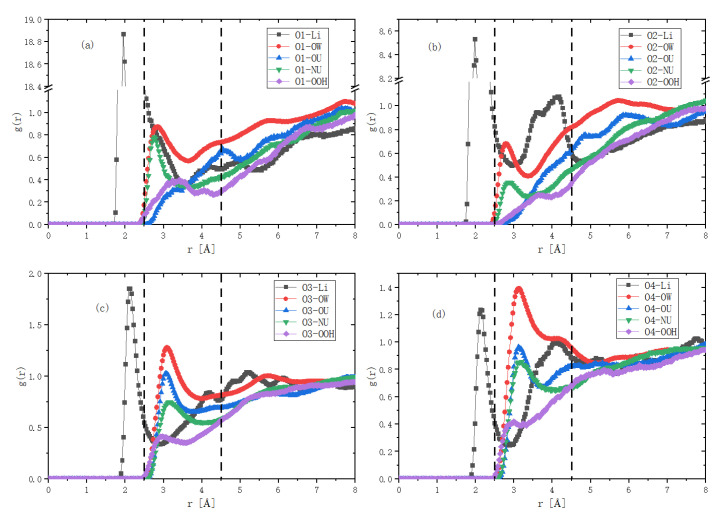
RDF distributions of some solvent atoms, i.e., Li, OW, OU, NU and OOH, around Os atoms in trehalose LiOH urea aqueous solution. (**a**–**d**) is the RDF distribution around O1 to O4, respectively. The dashed lines are used to indicate the boundary of the second hydration shells.

**Figure 6 molecules-27-03395-f006:**
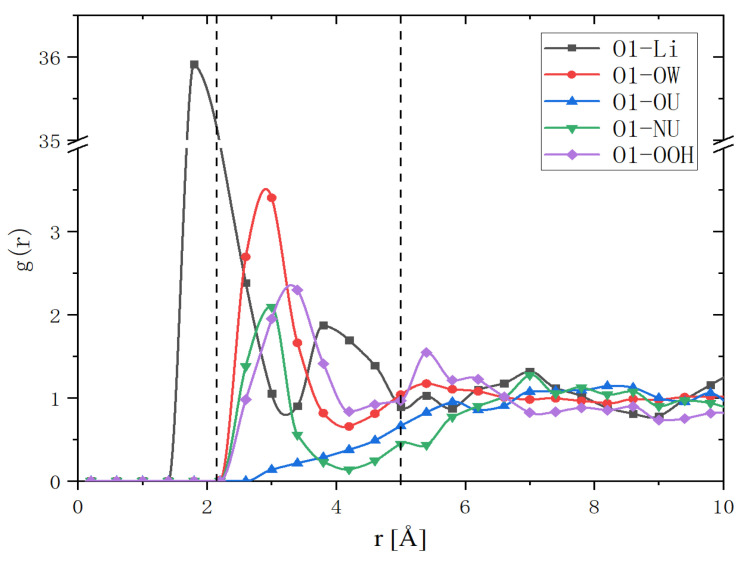
RDF distributions of Li, OW, OU, NU and OOH around O1 atom along *Z*-axis. The dashed lines are used to indicate the boundary of the second hydration layer.

**Figure 7 molecules-27-03395-f007:**
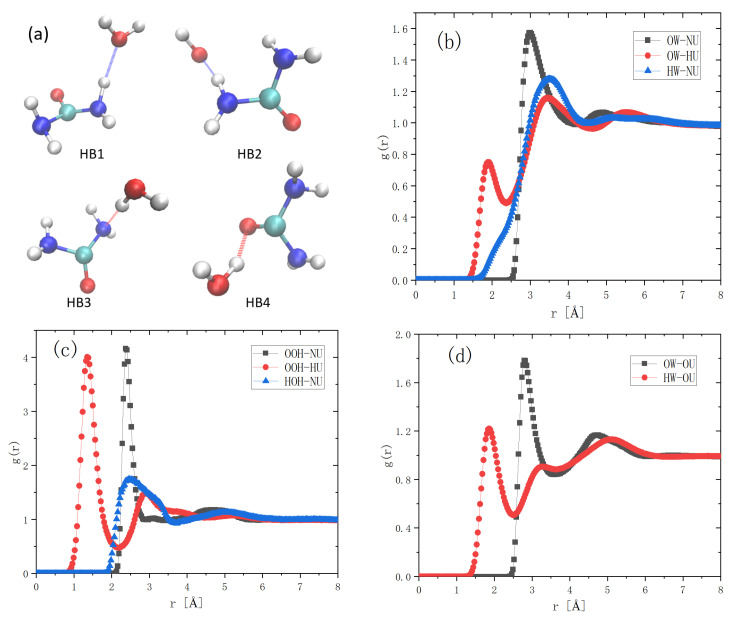
Schematic representation of the four important hydrogen bonds that urea forms with water and OH^−^ (**a**). The colors of the four atoms C, H, O and N are cyan, white, red and blue, respectively. RDF distributions between the atoms of urea and H_2_O/OH^−^ in trehalose LiOH urea aqueous solution (**b**–**d**).

**Figure 8 molecules-27-03395-f008:**
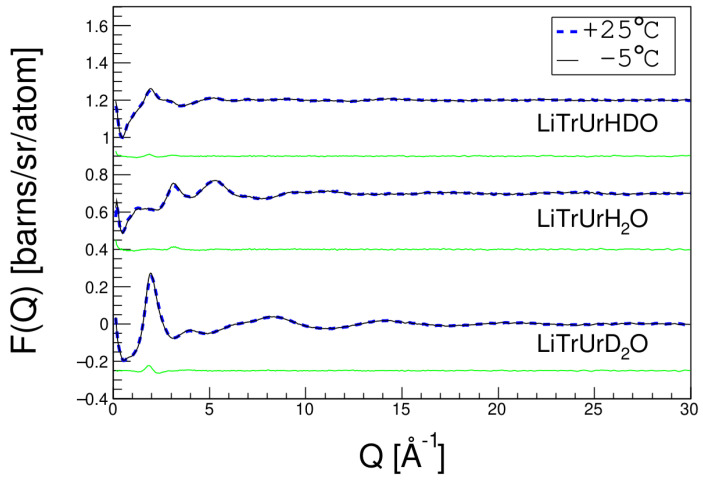
Neutron total scattering profiles of trehalose in LiOH/urea aqueous solution at 25 °C and −5 °C and the difference between them (green lines).

**Table 1 molecules-27-03395-t001:** Sample labels, chemical components, deuterium ratios and molar ratios of the samples.

Sample Labels	Li(K)TrH_2_O/HDO/D_2_O	Li(K)TrUrH_2_O/HDO/D_2_O
Chemical Component	Li(K)OH Trehalose Water	Li(K)OH Urea Trehalose Water
Deuterium Ratio	0.0/0.5/1.0	0.0/0.5/1.0
Molar Ratio	Li(K)OH:Urea:Trehalose:Water = 222:254:64:5716	

**Table 2 molecules-27-03395-t002:** Lennard-Jones (L-J) reference potential parameters and charge (q) of atoms used in EPSR.

**(1)**									
**Atom Label**	**OW**	**HW**	**Li**	**Na**	**K**	**OOH**	**HOH**	**CU**	**OU**
e [KJ/mole]	0.650	0.000	0.690	0.125	0.500	0.251	0.184	0.439	0.878
σ [Å]	3.166	0.000	1.510	2.500	3.000	2.750	1.443	0.375	2.960
q [e]	−0.848	0.424	0.679	0.679	0.679	−1.103	0.424	0.142	−0.390
**(2)**									
**Atom Label**	**NU**	**HU**	**C**	**O1**	**O2**	**O3**	**O4**	**H**	**M**
e [KJ/mole]	0.711	0.000	0.276	0.586	0.586	0.711	0.711	0.050	0.121
σ [Å]	3.250	0.000	3.500	3.100	2.900	3.100	3.100	1.700	1.700
q [e]	−0.542	0.333	0.258	−0.500	−0.500	−0.500	−0.500	0.301	0.000

**Table 3 molecules-27-03395-t003:** Cubic simulation box atomic number density and number of molecules used in EPSR.

Sample Labels	Li(K)OH	Urea	Trehalose	Water	Density (Atoms/Å^3^)
Li(K)TrH_2_O	222	0	64	5716	0.106531 (Li) 0.103323 (K)
Li(K)TrUrH_2_O	222	254	64	5716	0.106405 (Li) 0.103437 (K)

## Data Availability

The data that support the findings of this study are openly available in ISIS Neutron and Muon Source Data at http://doi.org/10.5286/ISIS.E.RB1820090 (accessed on 27 November 2018) and http://doi.org/10.5286/ISIS.E.RB1920190 (accessed on 15 December 2019).

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
