# Peer review of "Neutron Total Scattering Investigation of the Dissolution Mechanism of Trehalose in Alkali/Urea Aqueous Solution"

_molecules, 2022, doi:10.3390/molecules27113395_

Round 1
Reviewer 1 Report
The authors report the dissolution mechanism of cellulose in 4 alkali/urea/salt aqueous solutions using trehalose as a model of cellulose (trehalose/LiOH/urea, trehalose/KOH/urea, trehalose/LiOH and trehalose/KOH). The neutron scattering was used to study the atomic structure of liquids and empirical potential structure refinement simulation was used to explore the atom structure.
They demonstrated that ions, alkali, urea and water work together to dissolve trehalose and the formation of a hydration shield is crucial for dissolution.
The strengths of this work were to elucidate that Li+ ions are concentrated around O1 and O2, while K+ ions are more concentrated in the O3 and O4 region. This allowed to explain the reason why the aqueous solution of KOH urea at low temperature cannot dissolve cellulose, but it can dissolve chitin. The authors also suggest that the mechanism of cellulose dissolution is based on the breaking of the O1 and O2 hydrogen bond.
In my opinion, this research work is timely and presents relevant points for the entire academic society, as well as being of indisputable interest for readers of the Molecules Journal.
Minor errors
The quality (resolution) of images should be improved. For example, Figure 3 and Figure 9.
In Figure 3B it is not possible to see the black dots (urea atoms). Also has something written in non-English language (换种颜色)”
Author Response
Dear Reviewer,
The authors are grateful for the valuable comments and corrections made by the reviewers. We have listed all the specific changes and answers point by point according to comments, which are included in the following section. The corresponding changes in the re-submitted manuscript are highlighted, and through the review mode of MS Word, the comparison before and after modification is preserved.
Beat regards

Reviewer 2 Report
The manuscript employs neutron scattering experiments and Monte Carlo simulations to explain the mechanisms behind cellulose dissolution by alkali solution containing urea. This may be interesting to be published since cellulose dissolution has been widely studied in the last years. However, my great concern is related to the use of trehalose, a small, high-purity grade and water-soluble molecule to mimic cellulose. Authors should explain better how comparable these systems are, making sure that they are really comparable. Minor issues are also listed below. Based on this, I recommend its acceptance after major revisions.
- Introduction should be redesigned to better address the main issue to be investigated, based on the available literature, giving a clear picture of what has been already done for similar systems employing other techniques. The last paragraph of the introduction, for example, would be more appropriate for the conclusions.
- More details regarding the obtention of neutron scattering experiments, for example the acquisition time and the reference (blank) samples, should be provided in the experimental part.
- Experimental part states that the temperatures of 25o and -10oC have been employed, but results are displayed for 25o and -5oC. This should be corrected.
- Authors are suggested to check the works of Bjorn Lindman in this field.
Author Response
Dear Reviewer,
The authors are grateful for the valuable comments and corrections made by the reviewers. We have listed all the specific changes and answers point by point according to comments, which are included in the following section. The corresponding changes in the re-submitted manuscript are highlighted, and through the review mode of MS Word, the comparison before and after modification is preserved.
Best regards

Round 2
Reviewer 2 Report
Authors made all requested corrections. The manuscript is suitable for publication.